# Whole brain comparative anatomy using connectivity blueprints

Rogier B Mars[1,2]*, Stamatios N Sotiropoulos[1,3], Richard E Passingham[4], Jerome Sallet[4], Lennart Verhagen[4], Alexandre A Khrapitchev[5], Nicola Sibson[5], Saad Jbabdi[1]*

[1]Wellcome Centre for Integrative Neuroimaging, Centre for Functional MRI of the Brain (FMRIB), Nuffield Department of Clinical Neurosciences, John Radcliffe Hospital, University of Oxford, Oxford, United Kingdom; [2]Donders Institute for Brain, Cognition and Behaviour, Radboud University Nijmegen, Nijmegen, Netherlands; [3]Sir Peter Mansfield Imaging Centre, School of Medicine, University of Nottingham, Nottingham, United Kingdom; [4]Wellcome Centre for Integrative Neuroimaging, Department of Experimental Psychology, University of Oxford, Oxford, United Kingdom; [5]Cancer Research UK and Medical Research Council Oxford Institute for Radiation Oncology, Department of Oncology, University of Oxford, Oxford, United Kingdom

**Abstract** Comparing the brains of related species faces the challenges of establishing homologies whilst accommodating evolutionary specializations. Here we propose a general framework for understanding similarities and differences between the brains of primates. The approach uses white matter blueprints of the whole cortex based on a set of white matter tracts that can be anatomically matched across species. The blueprints provide a common reference space that allows us to navigate between brains of different species, identify homologous cortical areas, or to transform whole cortical maps from one species to the other. Specializations are cast within this framework as deviations between the species' blueprints. We illustrate how this approach can be used to compare human and macaque brains.
DOI: https://doi.org/10.7554/eLife.35237.001

*For correspondence:
rogier.mars@psy.ox.ac.uk (RBM);
saad@fmrib.ox.ac.uk (SJ)

**Competing interests:** The authors declare that no competing interests exist.

## Introduction

The ultimate goal of comparative and evolutionary neuroscience is to understand the organization of each species' brain as an adaptation to its unique ecological niche. However, the study of specific adaptations cannot be performed without an appreciation of the common organizational principles of different brains. To understand what is unique about the brain of a given species, a useful starting point is to cast it in the context of a common template. Unique properties and adaptations of a species' brain can then be understood as deviations from the template.

In higher primates, white matter organization has striking commonalities between the different species (*Thiebaut de Schotten et al., 2012*). Several association pathways have been identified in humans, chimpanzees, and macaques (*Rilling et al., 2008*; *Hecht et al., 2013*). These pathways share core properties such as the broad brain areas that they connect, but differ in the details of their branching patterns, suggesting a common connectivity backbone with varying degrees of connectivity specialization. We propose that common white matter pathways can be used to form *blueprints* of cortical connections to enable comparisons of cortical organization between higher primates.

We exploit the idea that cortical regions can be described by their unique sets of connections to the rest of the brain (*Passingham et al., 2002*), a feature that we have previously shown is useful in comparing brain organization between species (*Mars et al., 2016*). Thus, we can investigate neural organization using the architecture of the main white matter fibers. The bodies of the major fiber bundles can be identified reliably in different species and allow identification of homologous fiber bundles. This allowed us to construct a map of each of the main white matter tracts and to describe cortical grey matter organization in terms of this map (*Figure 1*). We term the matrix describing the connectivity of each vertex of the grey matter surface with each white matter tract the connectivity blueprint. These connectivity blueprints provide a common space in which we can ask how each part of the grey matter in one species maps onto the other species.

We illustrate this approach by comparing human and macaque cortex. We demonstrate that the connectivity blueprints can be used to predict the location of cortical areas across species. Furthermore, by quantifying the distances between the blueprints of different parts of the two brains, we quantify where these brains have tended to specialize since their last common ancestor. We demonstrate that such areas overlap with known specializations in the human and macaque lineages.

Our results show how connectivity blueprints can be used for comparative anatomy of humans and macaques, but the approach can be generalized to all higher primates where the blueprints can be identified. This method thus provides a powerful approach to comparative anatomy, and allows one to quantitatively define common principles and unique specializations in the brains of related animals.

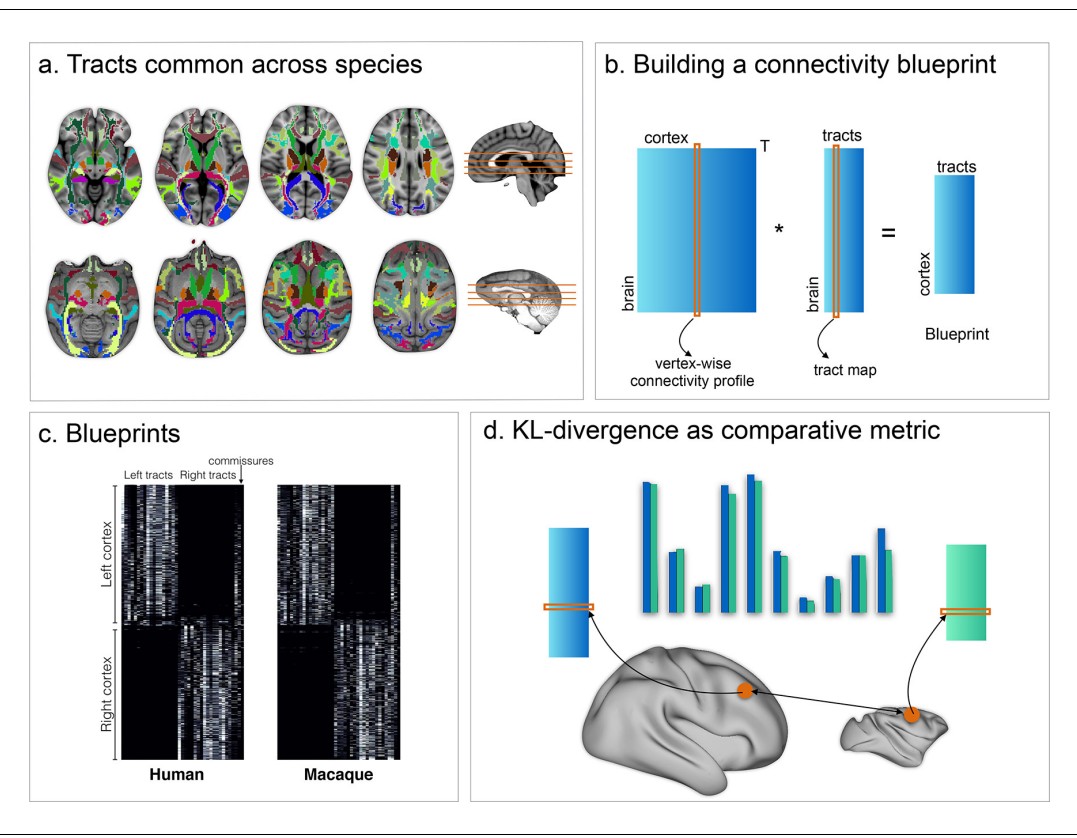

**Figure 1.** Methods overview. (**a**) 39 tracts common across both species were defined and reconstructed using probabilistic tractography. (**b**) The resulting connectivity matrices were then multiplied by connectivity matrices defining the connectivity of each vertex of the grey matter to the rest of the brain, creating a full connectivity blueprint (**c**) describing how each vertex is connected to each tract. (**d**) These blueprints can then be compared using the KL divergence as a comparative metric indicating how each vertex' connectivity fingerprint in one brain differs from that of each vertex in an other brain.

DOI: https://doi.org/10.7554/eLife.35237.002

## Results

### Comparing connectivity blueprints can identify homologous areas across brains

We first investigated whether the connectivity blueprints could be used to identify known homologs between the two species. Although the early visual areas are present in both humans and macaques, their location and the amount of cortical territory they occupy differs in the two species (*Orban et al., 2004*). A particularly challenging case is presented by areas sensitive to visual motion. The MT +complex is located in the ventrolateral part of the posterior temporal cortex in the human brain (hMT+; *Malikovic et al., 2016*), but is located more dorsally in the ventral bank of the posterior superior temporal sulcus in the macaque monkey (*Paxinos et al., 2000*) (*Figure 2*, left panel). hMT + can be identified as a region of high myelin in the posterior temporal cortex that can be visualized using the ratio of T1- and T2-weighted MRI scans (*Glasser and Van Essen, 2011*; *Large et al., 2016*). The peak of hMT + is reached by tracts associated with the visual system such as the occipital radiations and the ventral occipital fascicle, but also by longitudinal tracts such as the inferior longitudinal fascicle (*Yeterian and Pandya, 2010*). We created a map of the macaque cortex indicating how different each vertex's connectivity profile was to that of a vertex in hMT+. This map showed the lowest divergence, that is highest similarity, in the ventral bank of the macaque STS, as predicted from the macaque cytoarchitectonic atlas. Thus, comparison of connectivity blueprints can identify homologous areas across brains, even when their relative location has changed.

We next tested whether we could predict the location of the human pre-supplementary motor area (pre-SMA), based on macaque area F6. It has been well-established that these two regions share similar functions across the two species (*Nachev et al., 2008*) and can be matched based on their connectivity profiles (*Sallet et al., 2013*; *Mars et al., 2016*). Previous studies, however, matched the regions based on the profile of functional connectivity with known homologous brain regions in frontal and parietal cortex, rather than using white matter tracts that can potentially be identified in all higher primates. We defined macaque area F6 based on the atlas of *Markov et al. (2011)*. Its connectivity fingerprint shows that it receives widespread connections, including from the superior longitudinal fascicle, the cingulum bundle, and the frontal aslant (cf. *Thiebaut de Schotten et al., 2012*). We determined the Kullback-Leibler (KL) divergence between the connectivity fingerprint of F6 and that of each vertex of the human cortex. This map identified an area of the human medial prefrontal cortex, anterior to the supplementary motor area proper (*Figure 2*, right panel) and consistent with previous localizations of this area in the human (*Nachev et al., 2008*; *Mars et al., 2016*), as most similar to macaque area F6. This result demonstrates that matching connectivity blueprints across species can also be used to predict the location of areas outside early visual cortex.

### Connectivity blueprints can predict organization of the cortical surface across brains

As well as calculating divergence maps for a single vertex or a single area, the approach can be generalized to transform features of organization across the entire cortex between species. One such map that is easily obtainable from neuroimaging is a T1/T2-weighted map, which has been suggested to partly reflect the presence of cortical myelin (*Glasser et al., 2014*). T1/T2-weighted maps show a number of distinctive features across the human cortical hemisphere that are qualitatively similar to myelination maps, such as high values in primary sensory areas, low values in prefrontal and parietal association cortex, and intermediate values in frontal oculomotor areas. Using the connectivity blueprint as a reference space, we can transform a whole brain map from one species onto the other based on fingerprint similarities (see Materials and methods). We used this approach to predict the T1/T2-weighted map of the macaque cortex based on the same map in humans (*Figure 3*). The predicted map showed striking similarities to the actual macaque myelin map (*Glasser et al., 2014*), replicating the high myelin in the primary visual, auditory, and sensorimotor cortex and the low myelin in the prefrontal cortex.

There are also areas in which the predicted macaque T1/T2-weighted map differs from the actual map. For instance, the predicted map showed an intermediate level of myelin in the macaque inferior parietal cortex, whereas in reality this is an area with low myelin content. Thus, there are parts of

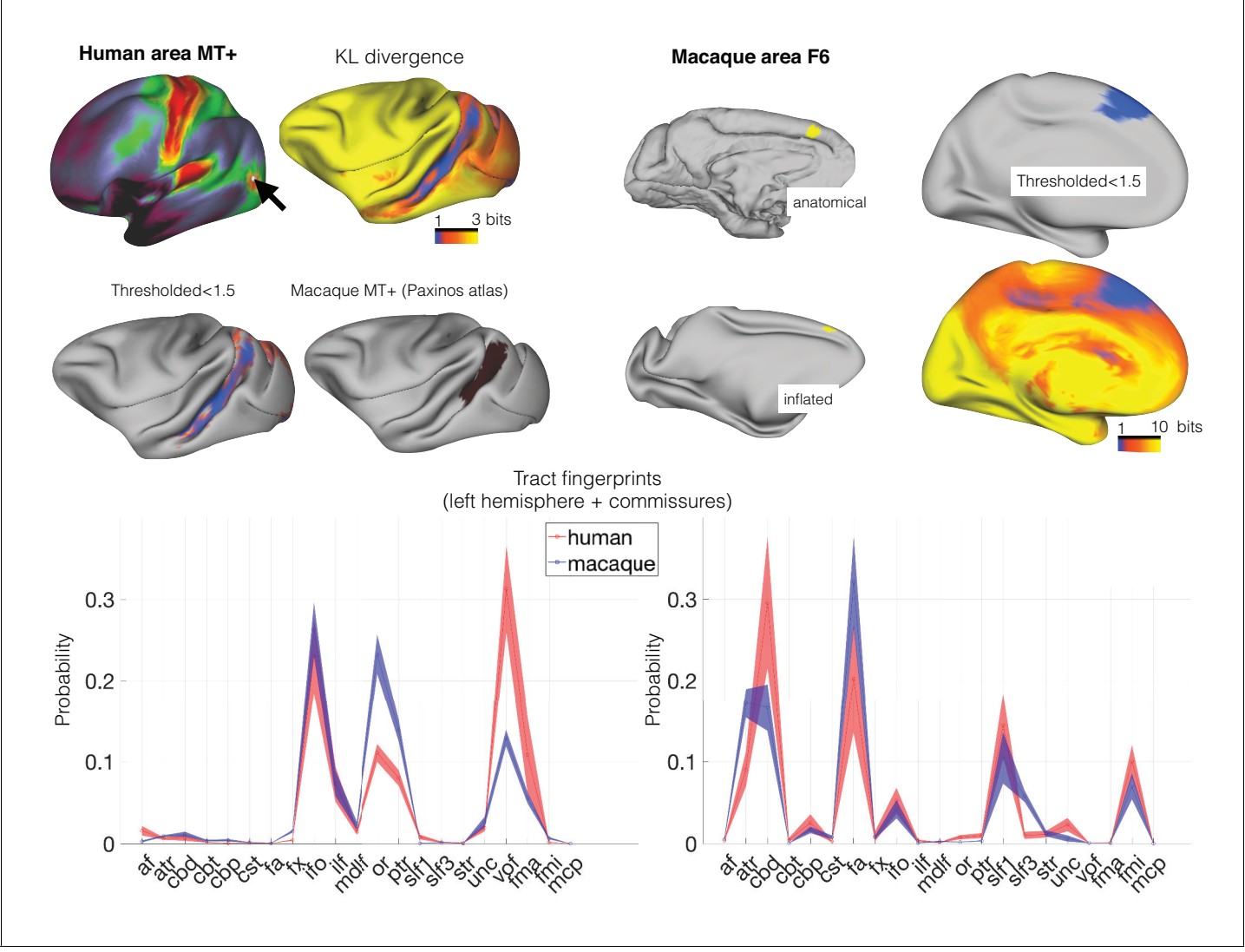

**Figure 2.** Identifying areas across species. (*left panel*) MT+ complex. Human MT+ can be defined as an area of high cortical myelin in the ventral occipitotemporal cortex (*top left*). Its connectivity fingerprint (shading indicates standard error) indicates strong projections from visual tracts such as the optic radiation (OR), vertical occipital fascicle (VOF), inferior fronto-occipital fascicle (IFO), and inferior longitudinal fascicle (ILF) (*bottom row*). According to previous work, macaque MT+ is located in the ventral bank of the superior temporal sulcus (*middle right*). Calculating the KL divergence of the connectivity fingerprint of human MT+ and the connectivity fingerprint of each macaque vertex (*top right*) shows the lowest divergence in the STS, with a thresholded image identifying the area predicted by previous work (*middle left*). (*right panel*) Area F6. Macaque area F6 (*left*) receives projections from, among others, the frontal aslant (FA) and the superior longitudinal fascicle (SLF) (*bottom row*). Calculating the KL divergence of the connectivity fingerprint of macaque F6 and the connectivity fingerprint of each human vertex shows the lowest divergence on the medial wall, with a thresholded image identifying human pre-SMA (*right*).

DOI: https://doi.org/10.7554/eLife.35237.003

the cortex whose organization we could not predict well based on the connectivity blueprint. While this could be due to limitations in the methods, it is noticeable that the poorer predictions are mostly located in the association cortex. These are areas whose organization might be unique to one of the two brains studied. We therefore sought to quantify dissimilarity in connectivity profiles between humans and macaques across the entire cortex.

## Connectivity blueprints identify unique aspects of brain organization

We investigated which parts of both the human and macaque brains are unique by creating a map of the distance of each vertex to its closest match in the other species. The greater the distance, the

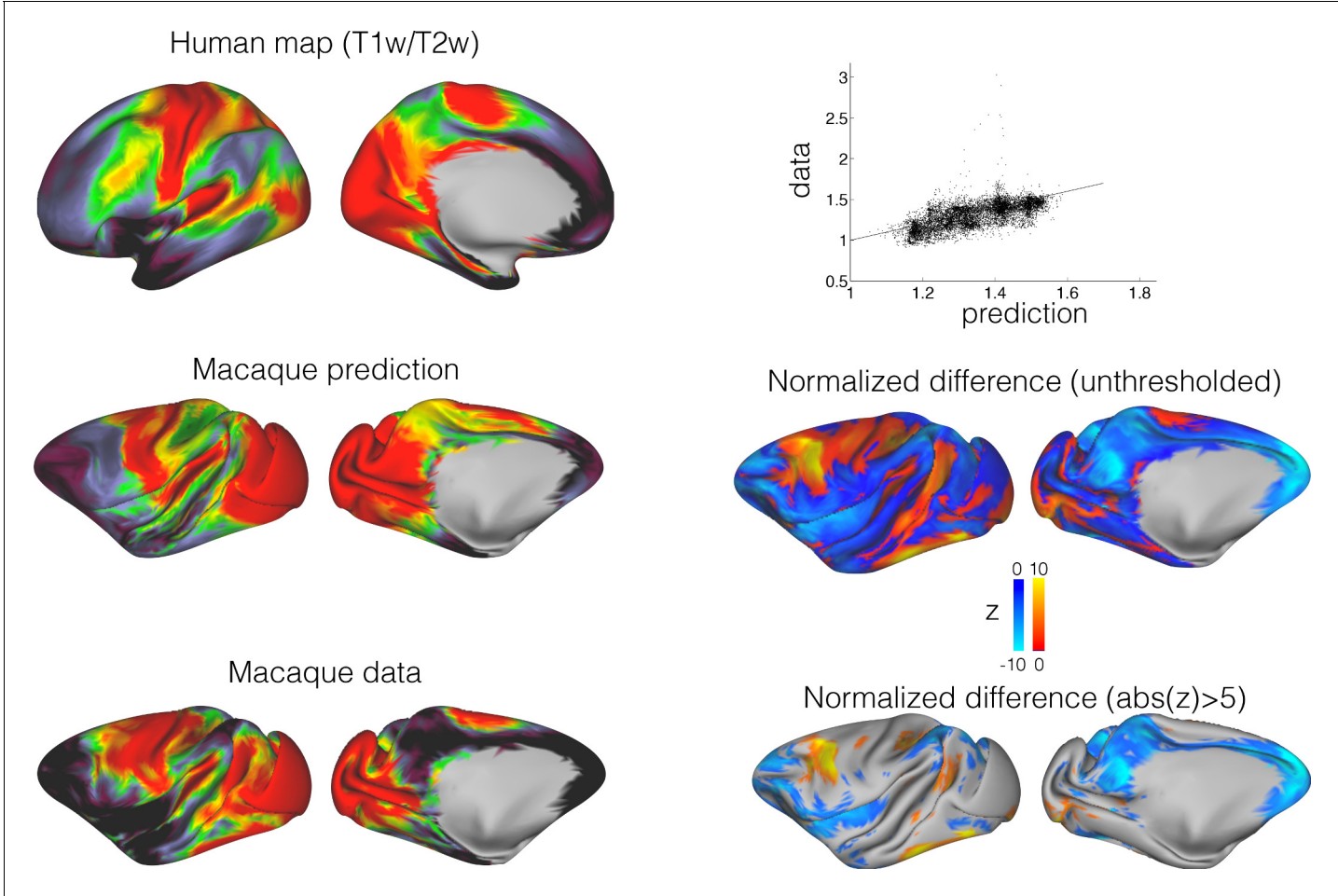

**Figure 3.** Predicting macaque T1/T2-weighted map from the human map. The connectivity blueprint can be applied to use the human T1/T2-weighted map (*top left*) and predict the same map in the macaque (*middle left*). The predicted macaque map shows strong similarity to an actual macaque map based on 19 macaques from the Yerkes dataset (*Donahue et al., 2018*). The scatter plot on the top right shows how well the predicted map follows the data (straight line is y = x). To assess the variability in the predicted map, we calculated a distribution based on individual variability (using all pairs of human/macaque datasets to build separate blueprints to drive the predictions). The resulting distribution was compared to the measured map (Z= (mean-data)/std) (middle right and bottom right). These assessments demonstrate that the predicted map shows striking similarities to the actual map, but important differences are noticeable in part of the association cortex.

DOI: https://doi.org/10.7554/eLife.35237.004

more likely this vertex has a connectivity profile that is not represented in the other species; in other words, the more likely this area has changed in its connectional organization since the last common ancestor of human and macaque. The resulting connectional dissimilarity map showed a large region of human inferior parietal and posterior temporal cortex, precuneus, and to a lesser extent parts of the frontal cortex that could not be predicted from any part of the macaque brain (*Figure 4*, top panel). Importantly, the between-species predictability did not correlate with any particular aspect of the connectivity fingerprint, such as a map of the entropy of tract distribution (i.e., whether a region is reached strongly by few tracts or equally strongly by multiple tracts) (*Figure 4*). Similarly, the connectional dissimilarity map overlapped with, but was different to a map of cortical expansion (*Van Essen and Dierker, 2007*), indicating that reorganization and expansion reflect separate aspects of brain reorganization (*Figure 4*).

The largest area in the human brain that showed a high connectional dissimilarity to the macaque is a section of inferior parietal and posterior temporal cortex. This section spans multiple cortical areas. We compared the fingerprint of the vertex with the highest minimum KL divergence in the human brain, that is, the vertex that has the least similar match in the macaque, to the fingerprint of

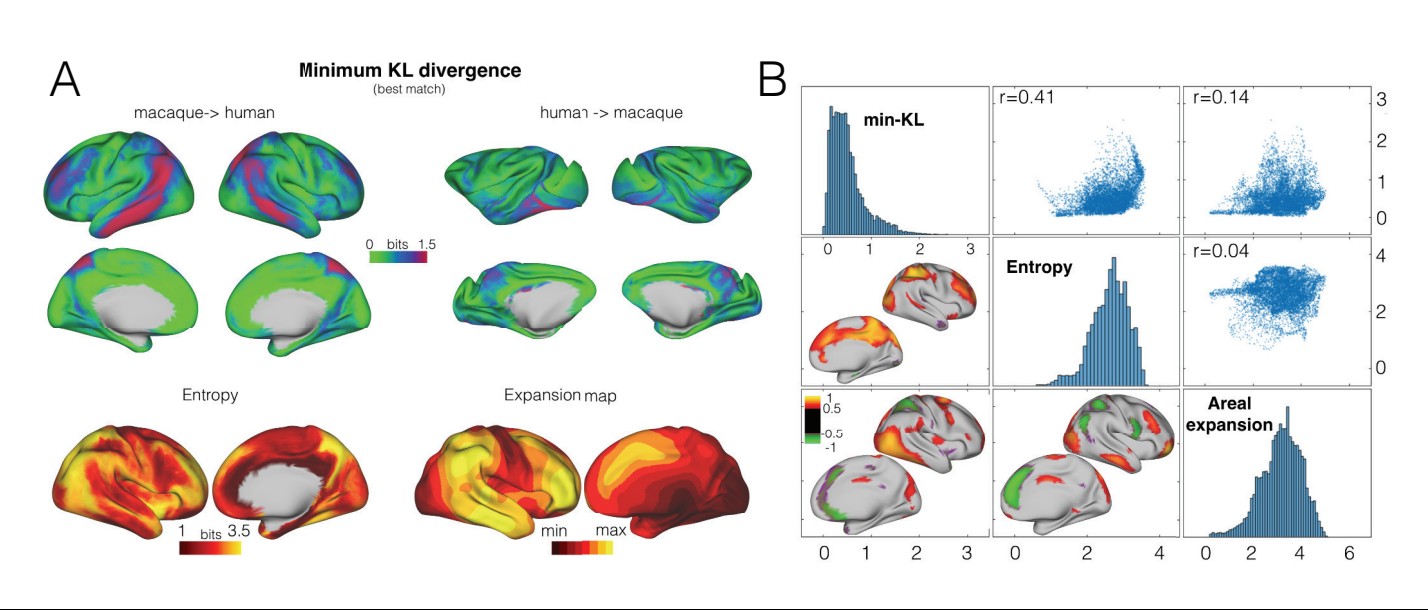

**Figure 4.** Divergence map. (A) A map of the minimum KL divergence of each vertex with all vertices in the other species brain indicates which areas are least similar across the two brain (*top panels*). This map can be compared with an entropy map showing the diversity of tracts reaching each vertex (*bottom left*) and a landmark-based cortical expansion map (figure generated from data available from the SUMSDB archive (http://brainvis.wustl.edu/sumsdb/archive_index.html) (*Van Essen and Dierker, 2007*) (*bottom right*). (B) We compared the KL divergence map to the entropy and expansion maps. Diagonal shows the distribution of values of each map, upper right scatter plots show relationship between all vertices of the pairs of maps; bottom left maps show the local correlations between two maps.

DOI: https://doi.org/10.7554/eLife.35237.005

the most similar macaque vertices (*Figure 5*). This showed that this vertex is reached very prominently by the arcuate fascicle (AF). The vertex is located in the posterior part of the temporal cortex, an area that often shows activation in phonological or semantic tasks (*Price, 2000*). Other parts of the cortex showing a high minimum divergence included the anterior part of the human angular gyrus. The angular gyrus has also been suggested to receive stronger AF connectivity than its proposed macaque homolog area PG (*Rilling et al., 2008*). This part of angular gyrus shows activation during phoneme detection (*Simon et al., 2002*) and has stronger grey matter density in bilinguals and adults who have learned to read compared to illiterates (*Carreiras et al., 2009*). Consistent with this role, neurons in macaque area PG show visual responses (*Rozzi et al., 2008*). Human angular gyrus receives input from the visual word form area (*Saygin et al., 2016*). Together, these results are consistent with the suggestion that the human brain contains areas with an organization not seen in the macaque in areas recruited into the language system.

Other human areas that have a connectivity fingerprint that was poorly predicted based on the macaque include the medial parietal cortex 7 m and areas in the lateral frontal cortex, including parts of dorsal prefrontal cortex. The medial parietal cortex is reached by the first branch of the superior longitudinal fascicle and this innervation seems stronger in the human brain. Based on shape analysis of structural imaging data of the human and chimpanzee, Bruner and colleagues have suggested that this area is preferentially expanded in the human brain (*Bruner et al., 2017*). The current results suggest that this expansion is accompanied by a change in connectivity. In the frontal cortex, the forceps minor of the corpus callosum seems stronger in the human than in the macaque, suggesting increased interhemispheric connectivity within the prefrontal cortex in this species.

## Translation of cortical atlases based on the connectivity blueprint

Another application of the blueprint approach to comparative anatomy is to use it to translate between brain atlases. Comparative atlases of different species' brains are rare in neuroscience, with most atlases focusing on a single species without explicit comparisons to others. The blueprint approach, however, can be used to translate between such different atlases. As an example we take

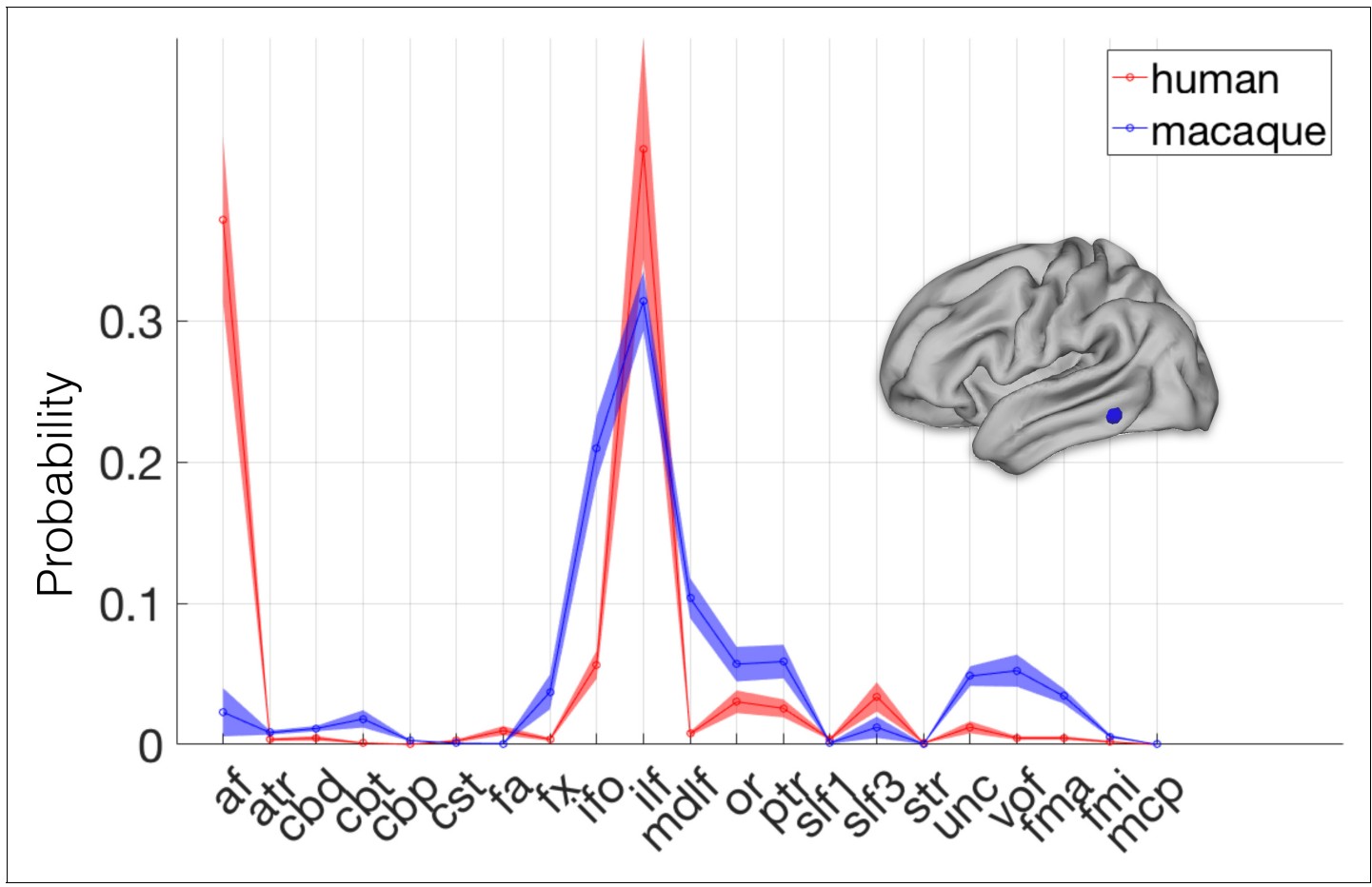

**Figure 5.** Connectivity fingerprint of an area with high divergence. The highlighted vertex on the human cortical surface has a connectivity fingerprint dissimilar to any found in the macaque. The vertex's connectivity fingerprint shows a much stronger influence of the arcuate fascicle (AF) even when compared to the most similar vertices in the macaque (the blue line is the average of the top 1% most similar macaque vertices). Shading indicates standard errors.

DOI: https://doi.org/10.7554/eLife.35237.006

the atlases of the human and vervet monkey cortex produced by Brodmann (**Brodmann, 1905**; **Brodmann, 1908**) which were converted to the human and macaque monkey surface in the Caret software (**Van Essen et al., 2012**). Brodmann labeled cytoarchitectonic areas in both species, but the labeling was not meant to indicate homologies (**Brodmann, 1909**; **Petrides et al., 2012**). We calculated the divergence between Brodmann areas in the two species and illustrated their similarities by projecting them to the same 2D space using spectral reordering (**Higham et al., 2007**) (**Figure 6**). At the gross level, this showed that regions within similar cortical systems group together in the 2D representation across the species. For instance, macaque primary visual areas 17 and 18 showed the smallest distance to human visual areas 17 and 18 and greatest to areas 24 and 25 belonging to the cingulate cortex and early sensorimotor areas 3 and 4 that do not receive any direct visual projections. Similarly, areas 23, 24, and 25, all reached by the cingulum bundle, tended to cluster together.

The fact that the nomenclature of Brodmann's maps is not always consistent between species is illustrated by monkey area 7 in the inferior parietal lobule (IPL). This area showed smallest dissimilarity to human area 40 rather than human area 7. This is consistent with the location of these areas, with human area 40 located on the angular gyrus of the IPL and human area 7 belonging to the superior parietal cortex (see highlighted area in **Figure 6**). This result confirms earlier suggestions that the IPLs in the two species are indeed most similar to one another (cf. **Mars et al. [2011]**) even though human IPL receives stronger arcuate connections, as demonstrated above. In sum, these

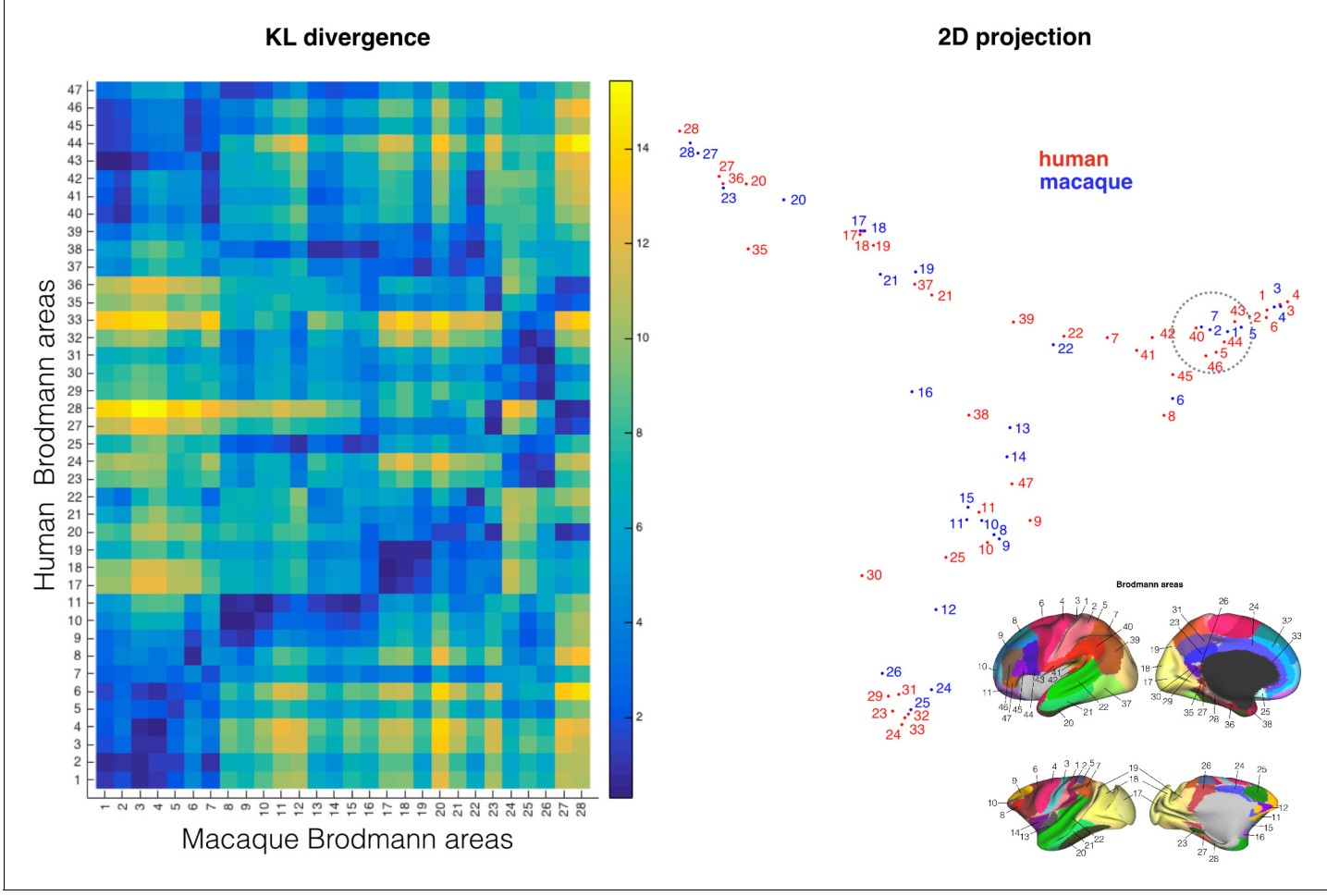

**Figure 6.** Comparing cortical atlases across species. The connectivity blueprints can be used to compare the connectivity fingerprint of cortical atlases of different species as illustrated here using Brodmann's maps of the human and monkey (*left*). Using spectral clustering, the divergence of the maps can be illustrated in a 2D representation, clustering together regions with the most similar connectivity fingerprint (*left*). Inset on the bottom right shows the atlas areas.

DOI: https://doi.org/10.7554/eLife.35237.007

results show that our approach can be used to translate existing cortical atlases across species, unifying previously diverse anatomical endeavors.

## Discussion

We have presented an approach to quantitatively compare cortical organization across species using their connectivity blueprint. We were able to predict known homologies between humans and macaques, such as the similarity of areas in visual cortex and the location of medial pre-SMA, but also to identify areas of diverging connectional reorganizations since the last common ancestor of humans and macaques. Moreover, we were able to provide a quantitative comparison of previously established atlases of the two species. We discuss the implications of this approach below, leaving a detailed discussion of its contribution to understanding macaque/human differences, including differences in lateralization and a further exploration of uniquely human aspects of temporal and frontal cortex organization, to a future communication.

Comparing the organization of the brains of related species is a fundamental challenge in neuroscience. Translational work relies on the assumption of evolutionary conservation, while evolutionary neuroscience aims to identify specializations explaining each species' unique adaptations (*Preuss et al., 2001*). Although cortical atlases are available for a number of model species, such as

the macaque and marmoset monkeys, these are often not built with explicit comparisons in mind (*Petrides et al., 2012*). Moreover, the laborious and invasive nature of traditional cortical mapping studies means there are few maps even of our closest relatives, such as the great apes. The current work exploits the benefits of neuroimaging to quickly acquire detailed anatomical data from whole brains, both in-vivo or based on post-mortem preserved tissue. The simplicity of this approach means it can be widely applied and easily extended.

The current results demonstrate which parts of the human cortex have a large connectional dissimilarity to the macaque. Earlier direct comparisons between the human and macaque cortex used surface-based registration based on a few known homologous cortical landmarks to create an expansion map showing which areas in the human brain have disproportionally expanded compared to the macaque (*Van Essen and Dierker, 2007*). This map showed areas of expansion in lateral prefrontal, inferior parietal, temporoparietal, and medial frontal cortex. Our connectional dissimilarity map is not an expansion map, but rather a map of connectional reorganization. Thus, the two maps describe separate aspects of cortical specialization, both of which are important in understanding what makes any one brain unique. Note, however, that the concept of the blueprint can also be used to create expansion maps that, rather than relying on morphological landmarks, use the blueprint as an anchor for measuring expansions. Similarly, a connectivity profile forms a different aspect of organization than the diversity of connections that a region receives, as indicated by our entropy map of tract distributions. There are different types of cortical organization that can result in a unique connectivity fingerprint, including invasion of new cortical territory as in the case of the arcuate and a change in the balance of connections due to strengthening of a particular connection (cf. *Mars et al., 2018*). A future step will be to create comparative maps that specifically quantify these different types of cortical reorganization.

Our approach to comparative anatomy effectively defines a common space, the connectivity blueprint, for the brains of different species based on connections with white matter tracts. This approach was chosen because the body of these tracts can be reliably identified, ensuring that the common space is based on properties that are homologous. The tracts were established using recipes developed by the authors. Although in agreement with the published literature, this inevitably requires some judgment calls. An alternative would be to describe the tracts based on observer-independent approaches (*O'Muircheartaigh and Jbabdi, 2018*), an approach that we aim to investigate in the future. However, if there is doubt regarding a particular tract, the current approach can also be used to test hypotheses regarding its course by testing the effect of various configurations on the similarity of the two brains. For example, one could search for a set of white matter tracts, a blueprint, that minimizes differences between cortical organization, under a parsimonious assumption of no connectional reorganization.

The ultimate strength in this approach is in its flexibility. It can be used to predict features of cortical organization such as the T1/T2-weighted map we have shown here, but also to predict how specific systems translate between species (e.g., the multiple demand network, *Mitchell et al., 2016*) or task-related activations when two species perform a similar task. Importantly, the approach can be generalized further by adapting the common space to include data from other modalities, including resting state functional MRI networks and maps of grey matter tissue properties such as myelin content or relative cortical thickness.

In summary, a connectivity blueprint approach to comparative anatomy can allow us to bridge cortical organizations in higher primates. Ultimately, this will lead to a reference template that represents common connectional organizations, deviations from which indicating species-specific specializations.

## Materials and methods

### Macaque data

Four post-mortem macaque diffusion MRI datasets were used. Data from one male macaque (*Macaca fascicularis*) from a previous study (*de Crespigny et al., 2005*) were obtained and preprocessed as described in *Jbabdi et al. (2013)*. Relevant imaging parameters were: 4.7T Oxford magnet equipped with BGA12 gradients; 3D segmented spin-echo EPI (430 um isotropic resolution,

eight shots, TE = 33 ms, TR 350 ms, 120 isotropically distributed diffusion directions, $b$-value = 8000 s/mm2.

Three additional macaque (*Macaca mulatta*) datasets (two male) were acquired locally on a 7T magnet with an Agilent DirectDrive console (Agilent Technologies, Santa Clara, CA, USA) using a 2D diffusion-weighted spin-echo protocol with single line readout (DW-SEMS, TE/TR: 25 ms/10 s; matrix size: 128 × 128; resolution: 0.6 x 0.6 mm; number of slices: 128; slice thickness: 0.6 mm). In these three monkeys, nine non-diffusion-weighted ($b = 0$ s/mm$^2$) and 131 diffusion-weighted ($b = 4000$ s/mm$^2$) volumes were acquired with diffusion directions distributed over the whole sphere. The brains were soaked in PBS before scanning and placed in fomblin during the scan. The $b = 0$ images were averaged and spatial signal inhomogeneities were restored. Diffusion-weighted images were processed using FMRIB's Diffusion Toolbox, first to fit diffusion tensors and estimate the mean diffusivity and fractional anisotropy, followed by voxel-wise model fitting of diffusion orientations using Bed-postX, using a crossing fiber model limited to three fiber directions (*Behrens et al., 2007*).

## Human data

Human in-vivo data was obtained from the minimally pre-processed data provided by the Human Connectome Project (www.humanconnectome.org) (*Van Essen et al., 2013*). All acquisition parameters and processing pipelines are described in detail in *Uğurbil et al. (2013)*, *Sotiropoulos et al. (2013)*, and *Glasser et al. (2013)*. The diffusion MRI data consisted of three shells (b-values = 1000, 2000, and 3000 s/mm$^2$) with 270 diffusion directions equally spread amongst the shells, and six b = 0 s/mm$^2$ acquisitions within each shell, with a spatial resolution of 1.25 mm isotropic voxels. Ten subjects were chosen randomly from the Q900 data release. Data were pre-processed with the HCP pipeline, which involves susceptibility-induced distortion correction (*Andersson et al., 2003*) and eddy-current distortion and motion correction (*Andersson and Sotiropoulos, 2016*). A crossing fibre model adapted to multi-shell data (*Jbabdi et al., 2012*) was fitted to the data prior to tractography.

## Surfaces

Models of the cortical surface were used for both humans and monkeys, including the pial surface and the white-gray matter interface. For humans, individual surface models were used, as provided through the HCP pipeline (*Glasser et al., 2013*), based on a Freesurfer surface reconstruction (*Dale et al., 1999*). For the macaque, we used surface reconstructions of one macaque with high quality structural MRI and nonlinearly (FSL's FNIRT) warped the other three macaque brains to enable using the same surface models in all four macaques. Macaque surfaces were then transformed to F99 standard space (*Van Essen, 2002*) to facilitate the combination of tractography results. All the surfaces (macaque and human) were downsampled from ~32 to ~10 k vertices prior to tractography analyses.

## Extracting the anatomical blueprint

Probabilistic diffusion tractography (*Behrens et al., 2007*) as implemented in FSL's probtrackx2 was used to extract the anatomical blueprints of macaques and humans. We extended an automated tractography tool (autoPtx, *De Groot et al., 2013*) to include a set of 39 major white matter bundles (18 on each hemisphere, and three cross-hemispheric pathways). Each bundle was reconstructed using a set of seed/inclusion/exclusion masks drawn in standard space (MNI152 for humans and F99 for macaques [*Van Essen, 2002*]). Tractography protocols for building the blueprints, code, and results are available for download from Gitlab at https://git.fmrib.ox.ac.uk/rmars/comparing-connectivity-blueprints.git (*Jbabdi et al., 2018*); copy archived at https://github.com/elifesciences-publications/rmars-comparing-connectivity-blueprints ).

## Creating connectivity blueprints

As shown in *Figure 1*, a connectivity blueprint consists of a (cortex) x (tracts) matrix where the tracts dimension is shared across both species. We build this matrix in two steps. First, we create a (cortex) x (whole brain) matrix by seeding probabilistic streamlines in standard space from every cortical vertex and recording the number of samples reaching each brain voxel (at 1 mm/2 mm resolution for F99 macaque/MNI152 human). This is done using the 'matrix2' mode in probtrackx2. Second, we

multiplied the resulting matrix with a (brain) x (tracts) matrix, thus creating a (cortex) x (tracts) matrix. Rows of this matrix can be interpreted (once normalized to sum to one) as the probability distribution of streamlines from a given vertex to connect to each of the 39 tracts.

## Comparing connectivity blueprints

We here introduce some mathematical notation: let M and H be the connectivity blueprint matrices for macaques and humans. For example, $M_{ik}$ quantifies the probability that vertex i in the macaque cortex connects to tract k. We normalize the rows of M and H to sum to 1, thus turning the rows into a discrete probability distribution.

To compare the fingerprint of vertex i in macaque to vertex j in humans, we use the symmetric Kullback-Leibler (KL) divergence (*Kullback and Leibler, 1951*) as a dissimilarity measure:

$$D_{ij} = \sum_k M_{ik} \log_2 \frac{M_{ik}}{H_{jk}} + \sum_k H_{jk} \log_2 \frac{H_{jk}}{M_{ik}}$$

Similarly, the same distance measure can be used to compare two vertices within species.

## Mapping between species

The similarity matrix calculated above can be used to transform a map from one species to the other using distance weighted interpolation (as done to map the myelin map from human to macaque in the Results section).

Given a map on the human cortex $h_i$ where i indexes vertices, we obtain a transformed macaque map *m as follows:*

$$m_j = \frac{\sum D_{ji}^{\gamma} h_i}{\sum D_{ji}^{\gamma}}$$

*where we used $\gamma = -4$.*

# Acknowledgements

This work was supported by the Biotechnology and Biological Sciences Research Council UK [BB/N019814/1]; the Netherlands Organization for Scientific Research NWO [452-13-015]; Cancer Research UK [C5255/A15935]; the Wellcome Trust [105651/Z/14/Z]; and the Medical Research Council UK [MR/L009013/1]. The Wellcome Centre for Integrative Neuroimaging is supported by core funding from the Wellcome Trust [203139/Z/16/Z]. RBM would like to thank Hiromasa Takemura for helpful discussion regarding VOF. The work leading to the macaque T1/T2-weighted 'myelin map' was supported in part by National Institutes of Health Grants P01AG026423 and the Yerkes National Primate Research Center base grant (Office of Research Infrastructure Programs; grant OD P51OD11132). We thank Nicole Eichert for help on aspects of surface processing.

# Additional information

## Funding

| Funder | Grant reference number | Author |
| --- | --- | --- |
| Biotechnology and Biological Sciences Research Council | BB/N019814/1 | Rogier B Mars |
| Wellcome | 203139/Z/16/Z | Rogier B Mars Jerome Sallet Saad Jbabdi |
| Nederlandse Organisatie voor Wetenschappelijk Onderzoek | 452-13-015 | Rogier B Mars |
| Cancer Research UK | C5255/A15935 | Alexandre A Khrapitchev Nicola Sibson |
| Medical Research Council | MR/L009013/1 | Saad Jbabdi |

The funders had no role in study design, data collection and interpretation, or the decision to submit the work for publication.

### Author contributions
Rogier B Mars, Conceptualization, Data curation, Formal analysis, Funding acquisition, Validation, Investigation, Methodology, Writing—original draft, Project administration; Stamatios N Sotiropoulos, Software, Methodology; Richard E Passingham, Supervision, Investigation, Writing—review and editing; Jerome Sallet, Data curation, Methodology; Lennart Verhagen, Data curation, Formal analysis, Methodology; Alexandre A Khrapitchev, Data curation; Nicola Sibson, Resources; Saad Jbabdi, Conceptualization, Software, Formal analysis, Supervision, Funding acquisition, Validation, Methodology, Writing—original draft

### Author ORCIDs
Rogier B Mars ![iD] http://orcid.org/0000-0001-6302-8631
Stamatios N Sotiropoulos ![iD] http://orcid.org/0000-0003-4735-5776
Jerome Sallet ![iD] http://orcid.org/0000-0002-7878-0209
Alexandre A Khrapitchev ![iD] http://orcid.org/0000-0002-7616-6635

### Ethics
Human subjects: Fully described in the core HCP literature referenced here; the paper is only using publicly available datasets on human imaging data.

### Decision letter and Author response
Decision letter https://doi.org/10.7554/eLife.35237.012
Author response https://doi.org/10.7554/eLife.35237.013

## Additional files

### Supplementary files
• Transparent reporting form
DOI: https://doi.org/10.7554/eLife.35237.008

### Data availability
The human diffusion MRI data was obtained from the Human Connectome Project (www.humanconnectome.org.). Tractography protocols for building the blueprints, code, and results are available for download from Gitlab at https://git.fmrib.ox.ac.uk/rmars/comparing-connectivity-blueprints.git.

The following previously published dataset was used:

| Author(s) | Year | Dataset title | Dataset URL | Database, license, and accessibility information |
|---|---|---|---|---|
| Van Essen DC, Smith SM, Barch DM, Behrens TE, Yacoub E, Ugurbil K, Consortium W--MH | 2013 | Human Connectome Project | www.humanconnectome.org | Publicly available at http://www.humanconnectome.org. |

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
