## [Decision Letter]

Thank you for submitting your article "Whole brain comparative anatomy using connectivity blueprints" for consideration by *eLife*. Your article has been reviewed by two peer reviewers, and the evaluation has been overseen by Klaas Stephan as Reviewing Editor and Sabine Kastner as the Senior Editor. The following individuals involved in review of your submission have agreed to reveal their identity: Michel Thiebaut de Schotten (Reviewer #1); Daniel S Margulies (Reviewer #2).

The reviewers have discussed the reviews with one another and the Reviewing Editor has drafted this decision to help you prepare a revised submission.

Summary:

This manuscript presents a novel approach to comparative anatomy across species, based on the idea of matching white matter tracts. This paper focuses on the comparison of human and macaque monkey brains. After identifying homologous tracts of the human and macaque monkey brains, these "connectivity blueprints" are used to define a common space for assessing how parts of grey matter in one species maps onto the other species. The correspondence across species is quantified using a dissimilarity measure from information theory, the Kullback-Leibler (KL) divergence.

Both reviewers and the handling editor found that the paper is original, innovative and of considerable importance for advancing the investigation of species differences and similarities in connectivity. We have a few suggestions for your consideration that may further improve the paper.

Essential revisions:

1) The authors implement a vertex-wise mapping to transform the human myelin map to the macaque monkey cortex, which they visually compare with a previously published map. While this analysis does establish proof-of-concept, and the results are indeed striking, it would be helpful to further present a direct statistical comparison of the two macaque monkey maps. For example, at this stage, the authors may want to consider how the question could already be assessed of where the species diverge based on the differences in myelin maps, and whether the localized divergence may be meaningful.

2) The whole-cortex comparison is the focus of the following analyses, which present the divergence maps across species, and visually compare them to previously published maps of connectivity entropy and cortical expansion based on landmarks. Again, while the visual comparison is certainly valid for providing a general impression of the overarching consistency and more subtle deviations across these approaches, any quantitative comparison of the maps that could help to reveal specific deviations would help further support the utility of the current approach based on connectivity correspondence. As the cortical maps presented in the macaque myelin and cortical expansion figures are both publicly available, this would be a feasible addition to the current manuscript, and would establish more rigorous comparison of different approaches to comparative cortical mapping.

---

## [Author Response]

Essential revisions:

1) The authors implement a vertex-wise mapping to transform the human myelin map to the macaque monkey cortex, which they visually compare with a previously published map. While this analysis does establish proof-of-concept, and the results are indeed striking, it would be helpful to further present a direct statistical comparison of the two macaque monkey maps. For example, at this stage, the authors may want to consider how the question could already be assessed of where the species diverge based on the differences in myelin maps, and whether the localized divergence may be meaningful.

Following the reviewer’s suggestion, we obtained the macaque myelin map from the Van Essen group and now provide a first statistical assessment between it and our predicted map. To do so, we exploit the individual variability to create a distribution of the predicted T1/T2-weighted map at each vertex (number of human subjects x number of macaque subjects predictions) and calculate a z-score indicating the normalized difference between the prediction and the actual map. These results have now been added to Figure 3:

“Figure 3. Predicting macaque T1/T2-weighted map from the human map. […] These assessments demonstrate that the predicted map shows striking similarities to the actual map, but important differences are noticeable in part of the association cortex.”

We have also added an acknowledgement for the use of the T1/T2-weighted macaque map:

“The work leading to the macaque T1/T2-weighted “myelin map” was supported in part by National Institutes of Health Grants P01AG026423 and the Yerkes National Primate Research Center base grant (Office of Research Infrastructure Programs; grant OD P51OD11132).”

Regarding the final point of the reviewers, whether the differences between the actual and predicted maps are meaningful, we refer to the next analyses, where we provide the full assessment of predictability across species. We make this link explicit in the manuscript:

“Thus, there are parts of the cortex whose organization we could not predict well based on the connectivity blueprint. […] We therefore sought to quantify dissimilarity in connections between humans and macaques across the entire cortex.”

In the spirit of using the variability across subjects as an indication of the reliability of our approach, we now have elected to add standard errors to all of the connectivity fingerprints shown in Figures 2 and 5.

2) The whole-cortex comparison is the focus of the following analyses, which present the divergence maps across species, and visually compare them to previously published maps of connectivity entropy and cortical expansion based on landmarks. Again, while the visual comparison is certainly valid for providing a general impression of the overarching consistency and more subtle deviations across these approaches, any quantitative comparison of the maps that could help to reveal specific deviations would help further support the utility of the current approach based on connectivity correspondence. As the cortical maps presented in the macaque myelin and cortical expansion figures are both publicly available, this would be a feasible addition to the current manuscript, and would establish more rigorous comparison of different approaches to comparative cortical mapping.

We have implemented the suggestion by the reviewer. We obtained the cortical expansion map from the Caret website and calculated the vertex-wise correlation between our divergence map and both the cortical expansion map and the entropy map (note that the entropy map is coming from this work and has not been published before). In addition, we calculated the local correlation cross the cortical surface between the different maps. The correlation maps have been added to Figure 4:

“Figure 4. Divergence map. (A) A map of the minimum KL divergence of each vertex with all vertices in the other species brain indicates which areas are least similar across the two brain (top panels). [...] Diagonal shows the distribution of values of each map, upper right scatter plots show relationship between all vertices of the pairs of maps; bottom left show the local correlations between two maps.”